# Regularized Gradient Clipping Provably Trains Wide and Deep Neural Networks

**Matteo Tucat**[*]                                          *matteotucat@gmail.com*
*Department of Computer Science*
*The University of Manchester*

**Anirbit Mukherjee**[†]                          *anirbit.mukherjee@manchester.ac.uk*
*Department of Computer Science*
*The University of Manchester*

**Mingfei Sun**                                    *mingfei.sun@manchester.ac.uk*
*Department of Computer Science*
*The University of Manchester*

**Procheta Sen**                                    *procheta.sen@liverpool.ac.uk*
*Department of Computer Science*
*University of Liverpool*

**Omar Rivasplata**                          *omar.rivasplata@manchester.ac.uk*
*Department of Computer Science*
*The University of Manchester*

**Reviewed on OpenReview:** *https://openreview.net/forum?id=ABT1XQLbOx*

## Abstract

We present and analyze a novel regularized form of the gradient clipping algorithm, proving that it converges to global minima of the loss surface of deep neural networks under the squared loss, provided that the layers are of sufficient width. The algorithm presented here, dubbed $\delta$-GClip, introduces a modification to gradient clipping that leads to a first-of-its-kind example of a step size scheduling for gradient descent that provably minimizes training losses of deep neural nets. We also present empirical evidence that this theoretically founded $\delta$-GClip algorithm is competitive with the state-of-the-art deep learning heuristics on various neural architectures including modern transformer based architectures. The modification we do to standard gradient clipping is designed to leverage the PL$^*$ condition, a variant of the Polyak-Łojasiewicz inequality which was recently proven to be true for sufficiently wide neural networks at any depth within a neighbourhood of the initialization.

## 1 Introduction

In various disciplines, from control theory to machine learning theory, there has been a long history of trying to understand the nature of convergence on non-convex objectives for first-order optimization algorithms, i.e. algorithms which only have access to (an estimate of) the gradient of the objective (Maryak & Chin, 2001; Fang et al., 1997). The new incarnation of this question in optimization problems in high dimensions, which arise in modern machine learning applications including neural network training, motivate the need for finite-time analysis of such algorithms. However, a challenging aspect of these modern use cases is their reliance on fine-tuning of some hyper-parameters, such as the step size, momentum, and batch size. In the wake of this, the "adaptive gradient" algorithms, such as Adam (Kingma & Ba, 2014) have become essentially indispensable for deep learning (Sun & Spall, 2019; Melis et al., 2018; Bahar et al., 2017).

---

[*]Work done while a student at the Department of Computer Science, University of Manchester.
[†]Corresponding author.

The widespread popularity of adaptive gradient methods in deep learning, like Adam, arguably stems from the fact that it seems easy to find task-specific and useful neural nets for which the default settings of these algorithms work well out of the box. Adam-like methods use as their update direction a vector which is the image of a linear combination of some (or all) of the gradients seen until the current iterate, under a linear transformation—often called the "diagonal pre-conditioner"—constructed out of the history of the gradients. It is generally believed that this "pre-conditioning" makes these algorithms much less sensitive to the selection of its hyper-parameters. An important precursor to Adam was the AdaGrad algorithm (Duchi et al., 2011). The far-reaching usefulness of adaptive gradient methods has motivated significant attempts at their theoretical justifications in the non-convex setting.

However, to the best of our knowledge, there has not been so far a theoretical guarantee for any adaptive gradient algorithm to converge to the global minima of deep neural network loss surfaces.

On the other hand, in recent times a number of motivations have come to light to consider training algorithms beyond these conventional adaptive gradient algorithms (Bernstein et al., 2018). In works like Simsekli et al. (2019) and Zhang et al. (2020b), a number of reasons have been pointed out as to how gradient clipping based adaptivity is better suited for deep learning. In this kind of adaptivity, the primary interest is on mechanisms to prevent the algorithm from using arbitrarily large gradients. Gradient clipping has been successfully deployed in a wide range of problems, particularly in natural language processing tasks such as GPTs (Brown et al., 2020) and LSTMs (Merity et al., 2018), and more recently in computer vision tasks (Brock et al., 2021). Clipping the gradient is also known to alleviate the problem of exploding gradients in recurrent neural networks (Pascanu et al., 2012; 2013), as well as helping to provide privacy guarantees in differentially private machine learning (Abadi et al., 2016; Ma et al., 2023). However, as for Adam, gradient clipping, too, has no known theoretical guarantee of training neural nets.

Inspired by the above, in this work we introduce a form of gradient clipping which in experiments we demonstrate to be competitive with Adam, vanilla stochastic gradient descent (SGD) and standard gradient clipping, while also being guaranteed to train neural nets of arbitrary depth when training on the squared loss and when the layers are sufficiently wide. Our proof crucially leverages the novel PL* condition that was proven for such nets in Liu et al. (2022). *Thus, we give a first-of-its-kind deep learning algorithm that is of practical benefit while being rigorously provable to minimize loss functions of deep neural nets.*

**Summary of Results**

In Zhang et al. (2020a), the following specific form of gradient clipping (which from here onwards we will refer to as "standard gradient clipping" or "GClip") is studied.

**Definition 1 (GClip).** For any $\eta, \gamma > 0$, the standard gradient clipping (GClip) algorithm for a differentiable objective function $f$ is defined as

$$\boldsymbol{x}_{t+1} = \boldsymbol{x}_t - h(\boldsymbol{x}_t) \cdot \nabla f(\boldsymbol{x}_t),$$

$$\text{with} \quad h(\boldsymbol{x}_t) := \eta \cdot \min\left\{1, \frac{\gamma}{\|\nabla f(\boldsymbol{x}_t)\|}\right\}.$$

The term $\gamma$ acts as the gradient norm threshold. To the best of our knowledge, this algorithm has no known convergence guarantees for deep learning, motivating us to present a modification of GClip, which we refer to as $\delta$-Regularized-GClip, or $\delta$-GClip for short.

**Definition 2 ($\delta$-Regularized-GClip).** For $\delta \in (0, 1)$ and $\eta, \gamma > 0$, the $\delta$-Regularized-GClip ($\delta$-GClip) algorithm for a differentiable objective function $f$ is defined as

$$\boldsymbol{x}_{t+1} = \boldsymbol{x}_t - h(\boldsymbol{x}_t) \cdot \nabla f(\boldsymbol{x}_t),$$

$$\text{with} \quad h(\boldsymbol{x}_t) := \eta \cdot \min\left\{1, \max\left\{\delta, \frac{\gamma}{\|\nabla f(\boldsymbol{x}_t)\|}\right\}\right\}.$$

The critical $\max\{\delta, ...\}$ term ensures $h(\boldsymbol{x}_t) \geq \eta\delta$, thus preventing $h(\boldsymbol{x}_t)$ from vanishing as $\|\nabla f(\boldsymbol{x})\| \to \infty$. It is important to note that, due to this modification, the distance between any two iterates $\|\boldsymbol{x}_{t+1} - \boldsymbol{x}_t\|$ is not bounded as the gradient norm grows.

Written alternatively, $\delta$-GClip for $\delta \in (0,1)$ implements GD with a gradient norm dependent step-length i.e

$$\boldsymbol{x}_{t+1} = \boldsymbol{x}_t - h_t \cdot \nabla f(\boldsymbol{x}_t)$$

$$\text{with} \quad h_t = \begin{cases} \eta, & \text{if } \|\nabla f(\boldsymbol{x}_t)\| \leq \gamma \\ \frac{\eta\gamma}{\|\nabla f(\boldsymbol{x}_t)\|} & \text{if } \gamma < \|\nabla f(\boldsymbol{x}_t)\| \leq \frac{\gamma}{\delta} \\ \eta\delta & \text{if } \|\nabla f(\boldsymbol{x}_t)\| > \frac{\gamma}{\delta} \end{cases} . \tag{1}$$

Note that setting $\delta = 0$ in the definition of $\delta$-GClip above recovers standard GClip (Definition 1). And at $\delta = 1$, $\delta$-GClip is the standard GD at a constant step-length. Thus $\delta$-GClip can be viewed as a particular choice of continuously interpolating algorithms between gradient clipping and gradient descent — and we will prove and demonstrate that $\delta \in (0,1)$ provides for wanted deep-learning properties not known to be obtainable at either extremities.

In the experiments given in Section 3 we shall see that, in practice, $\delta$ has to be chosen very small; though its presence is critical for the convergence guarantee that we shall establish in Theorem 2.1, which is the main theoretical contribution for our algorithm $\delta$-GClip.

The informal description of our key result is as follows: *Given a deep neural network that is sufficiently wide (parametric in $\delta$), $\delta$-GClip will minimize the squared loss to find a zero-loss solution at an exponential convergence rate, for any training data.* To the best of our knowledge, $\delta$-GClip is the first instance of an adaptive gradient algorithm or GD with a non-trivial learning rate schedule, that provably minimizes the empirical loss of neural nets at any depth. Additionally, our experiments show that $\delta$-GClip offers competitive performance when compared against state-of-the-art deep learning optimizers — for architectures (like transformers) and losses far outside the ambit of the proof given.

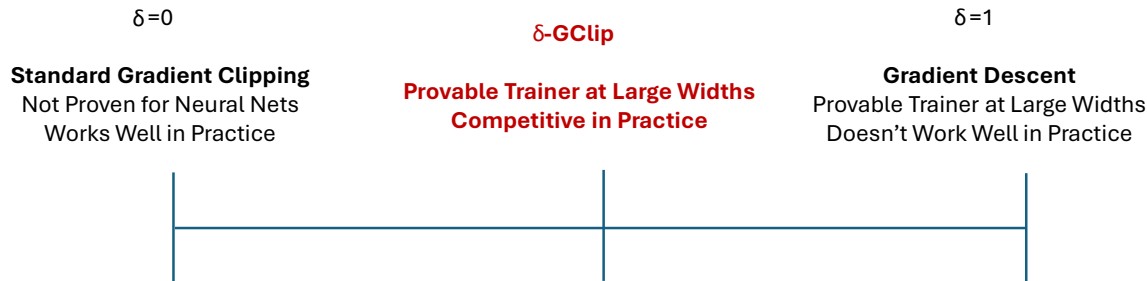

Figure 1: A Summarized View of $\delta$-GClip.

A stochastic version of $\delta$-GClip will also be considered in Section 2 (cf. Definition 5), which is relevant when under a certain noisy gradient setup, and we state a convergence result for it in Theorem 2.2.

**Organization**

In Section 2 we give the formal statements of our convergence theorems for our adaptive gradient method, $\delta$-GClip. In Section 3 we demonstrate empirically that our algorithm can compete with various popular deep learning heuristics on benchmark tests with a VAE, ResNet-18, Vision Transformer and a BERT based model. In Section 4 we review the current literature on provable deep learning optimization algorithms, focusing on adaptive gradient algorithms. In Section 5 we give the full proof of our main theorem on provable minimization of the squared loss for deep nets by our $\delta$-GClip when using full gradients. We conclude, in Section 6, discussing possible future directions of research. Appendix A contains the proof of convergence of stochastic $\delta$-GClip.

**Notation** The Euclidean ball centered at $\boldsymbol{w}_0 \in \mathbb{R}^m$ of radius $R$ is $B(\boldsymbol{w}_0, R) := \{\boldsymbol{w} \in \mathbb{R}^m : \|\boldsymbol{w}_0 - \boldsymbol{w}\|_2 \leq R\}$. Unless otherwise stated, $\|.\|$ denotes the $\ell_2$-norm for vectors and the spectral norm for matrices.

## 2  Theory for $\delta$-Regularized-GClip

This section contains theory results for $\delta$-GClip (Theorem 2.1) and for its stochastic version (Theorem 2.2). Towards the theory results, we first outline technical definitions.

**Definition 3** ($\mu$-PL$^*$ Condition). A non-negative loss function $\mathcal{L}$ is said to satisfy $\mu$-PL$^*$ on a set $\mathcal{S} \subset \mathbb{R}^m$ if $\exists \mu > 0$ such that $\forall \boldsymbol{w} \in \mathcal{S} : \|\nabla\mathcal{L}(\boldsymbol{w})\|^2 \geq \mu\mathcal{L}(\boldsymbol{w})$.

Next, the $L$-hidden layer feed-forward neural network architectures are recalled, and their loss setups, which were within the ambit of considerations in Liu et al. (2022).

**Definition 4.** A neural network is given by

$$f(\boldsymbol{w}; \boldsymbol{x}) = \alpha^{(L+1)}, \quad \alpha^{(0)} = \boldsymbol{x}, \quad \alpha^{(l)} = \sigma_l\left(\frac{1}{\sqrt{m_{l-1}}} \cdot W^{(l)}\alpha^{(l-1)}\right) \quad \text{for } l \in [1, L+1],$$

where $W^{(l)} \in \mathbb{R}^{m_l \times m_{l-1}}$ is the matrix of connection weights for the $l$-th layer, $m_l$ is the width of the $l$th layer with $m_{L+1} = 1$, and $\alpha^{(l)}$ is the output from the $l$-th layer, after the activation $\sigma_l$ is applied. Also, the 'weight vector' $\boldsymbol{w}$ encompasses all weights across all layers. We assume that the last layer activation $\sigma_{L+1}$ is $L_\sigma$-Lipschitz continuous and $\beta_\sigma$-Lipschitz smooth ($\beta_\sigma$-smooth), and satisfies $|\sigma'_{L+1}(\boldsymbol{z})| \geq \rho > 0$.

We train the weights of $f(\boldsymbol{w}, \cdot)$ using an $n$-sample training dataset, $\{\boldsymbol{z}_i = (\boldsymbol{x}_i, y_i) \mid i = 1, ..., n\}$. We write $\mathcal{F}(\boldsymbol{w}) = (f(\boldsymbol{w}; \boldsymbol{x}_1), ..., f(\boldsymbol{w}; \boldsymbol{x}_n)) \in \mathbb{R}^n$ for the vector of outputs for all training samples, and $\boldsymbol{y} = (y_1, \ldots, y_n)$. We use the squared loss $\mathcal{L}(\boldsymbol{w}) = \frac{1}{2}\|\mathcal{F}(\boldsymbol{w}) - \boldsymbol{y}\|^2$.

Now we have all the requisite background to state the key theorem pertaining our algorithm $\delta$-GClip.

**Theorem 2.1** ($\delta$-Regularized-GClip Provably Trains Wide and Deep Neural Nets). Suppose an overparametrized neural network $f$ is being trained using the square loss $\mathcal{L}(\boldsymbol{w})$, as specified in Definition 4. Then $\exists \ \lambda_0 > 0$ s.t for any $\eta, \mu, \delta > 0$ appropriately small enough and $\delta < 1$, if the minimum width of the network layers satisfies,

$$m = \tilde{\Omega}\left(\frac{nR^{6L+2}}{(\lambda_0 - \mu\rho^{-2})^2}\right) \quad \text{with} \quad R = \frac{\eta\sqrt{2\beta}\sqrt{\mathcal{L}(\boldsymbol{w}_0)}}{1 - \sqrt{1 - \frac{1}{2} \cdot \eta\delta\mu}}, \tag{2}$$

then one can initialize the weights s.t, w.h.p over initialization, the above loss is $\mu$-PL$^*$ in the ball $\mathrm{B}(\boldsymbol{w}_0, R)$ around initialization $\boldsymbol{w}_0$. Furthermore, let $\beta_\mathcal{F}$ be s.t $\mathcal{F}(\boldsymbol{w})$ is locally $\beta_\mathcal{F}$-smooth in $\mathrm{B}(\boldsymbol{w}_0, R)$. Then, training such a network using $\delta$-Regularized-GClip with $\eta < \min\{\frac{1}{\beta_\mathcal{F}}, \frac{1}{\mu}\}$ results in geometric convergence to a global minimizer $\boldsymbol{w}_* \in \mathrm{B}(\boldsymbol{w}_0, R)$ such that $\mathcal{L}(\boldsymbol{w}_*) = 0$ and at a rate given as,

$$\mathcal{L}(\boldsymbol{w}_t) \leq \mathcal{L}(\boldsymbol{w}_0)\left(1 - \frac{1}{2} \cdot \eta\delta\mu\right)^t. \tag{3}$$

**Remark.** The assumptions of $\eta < 1/\mu$ and $\delta < 1$ imply $\left(1 - \frac{1}{2} \cdot \eta\delta\mu\right) \in \left(\frac{1}{2}, 1\right)$, hence $\lim_{t\to\infty} \mathcal{L}(\boldsymbol{w}_t) = 0$.

This theorem fulfils the theoretical guarantee for our $\delta$-GClip algorithm. The proof is deferred to Section 5.

Next, we consider a stochastic version of our $\delta$-GClip algorithm, defined as follows.

**Definition 5** (Stochastic $\delta$-Regularized-GClip). We define the stochastic $\delta$-Regularized-GClip algorithm for a differentiable function $\mathcal{L}$ as

$$\boldsymbol{w}_{t+1} = \boldsymbol{w}_t - h(\boldsymbol{g}_t) \cdot \boldsymbol{g}_t,$$

$$\text{where} \quad h(\boldsymbol{g}_t) = \eta\min\left\{1, \max\left\{\delta, \frac{\gamma}{\|\boldsymbol{g}_t\|}\right\}\right\}$$

$$\text{and} \quad \mathbb{E}[\boldsymbol{g}_t \mid \boldsymbol{w}_t] = \nabla\mathcal{L}(\boldsymbol{w}_t) \tag{4}$$

for any $\eta, \gamma > 0$, $\delta \in (0, 1)$, and an arbitrary choice of $\boldsymbol{w}_1$, the initial point.

Towards analyzing the stochastic version of $\delta$-GClip just defined, we make the following assumptions.

**Assumption 1.** $\exists\ \theta \geq 0$ s.t. $\forall \boldsymbol{w},\ \|\boldsymbol{g}(\boldsymbol{w}) - \nabla\mathcal{L}(\boldsymbol{w})\| \leq \theta$.

**Assumption 2.** $\mathcal{L}$ is non-negatively lower bounded i.e. $\min_{\boldsymbol{w}}\mathcal{L} = \mathcal{L}_* \geq 0$.

**Assumption 3.** $\mathcal{L}$ is $\beta$-smooth.

The following theorem holds for stochastic $\delta$-GClip.

**Theorem 2.2** (**Convergence of Stochastic $\delta$-Regularized-GClip**)**.** Given Assumptions 1, 2 and 3, and for an arbitrary choice of $\epsilon > 0$, let $\epsilon' := \epsilon/\theta$. Then, for $\beta = 1$, $\delta = (1 + 2\epsilon'^2)/(1 + 3\epsilon'^2)$, and $\eta = (\frac{1}{4} \cdot \epsilon'^2)/(1 + \epsilon'^2)$, stochastic $\delta$-Regularized-GClip iterates satisfy the following inequality:

$$\text{for}\ \ T \geq \frac{\theta^4}{\epsilon^4}, \qquad \min_{t=1,\dots,T}\mathbb{E}\left[\|\nabla\mathcal{L}(\boldsymbol{w}_t)\|^2\right] \leq \mathcal{O}(\epsilon^2).$$

The proof of this theorem is given in Appendix A, where we first prove a slightly more general result.

We note that albeit $\delta$-GClip only partially clips the gradient, the convergence guarantee above does not need the gradient norms to be bounded as was also the case for standard stochastic gradient clipping, cf. Theorem 7 in Zhang et al. (2020a). *Next,* we note that the convergence guarantee for standard stochastic clipping does not immediately hold as stated in Zhang et al. (2020a) for the standard smoothness assumption that is used here. *Lastly,* unlike standard gradient clipping, the work presented here offers convergence guarantees in the deterministic ("full gradient") setting (Theorem 2.1) as well as the stochastic ("noisy gradient") setting (Theorem 2.2), for the same clipping algorithm.

# 3 Experimental Evidence for The Performance of $\delta$-Regularized-GClip

This section contains experimental evidence for $\delta$-GClip. We split our experimental demonstrations into two segments. In the next subsection we focus on certain conventional deep learning models and in the later subsection we focus on transformer based models. The experiments cover text as well as image data. We note that all the models considered here are outside the scope of the theory proven previously and hence the success of $\delta$-GClip in such varied scenarios can be seen to robustly demonstrate its practical abilities as a deep learning algorithm. The code for all our experiments can be found in the GitHub repository.[1]

## 3.1 Experiments for $\delta$-Regularized-GClip on a ResNet and a VAE

We demonstrate here that the regularization term in $\delta$-Regularized-GClip helps improve the performance of standard gradient clipping—which anyway outperforms stochastic gradient descent (SGD)—and is in fact competitive when compared against the most popular optimizers such as Adam, even surpassing it in some cases. We test in supervised classification as well as unsupervised distribution learning settings.

We perform four sets of experiments. The first set is on the standard ResNet-18 (He et al., 2016) being trained on the benchmark CIFAR-10 (Krizhevsky, 2009) dataset, which we recall is a 10-class image classification task with $50,000$ training images and $10,000$ test images. The second set of experiments is training a VAE model on the Fashion-MNIST dataset, with $60,000$ training samples and $10,000$ for testing. Further, each of these is done both with learning rate scheduling—whereby the learning rate $\eta$ is reduced at certain points in the training—and without such scheduling (constant $\eta$ throughout).

In the (supervised) classification experiments, the training is done using the cross-entropy loss and ReLU gate nets, and using weight-decay (of 5e−4). Whereas the VAE setup does not have a loss function in the same conventional sense as considered in the theorems in Section 2. Hence, these experiments demonstrate the efficacy of regularized gradient clipping ($\delta$-GClip) beyond the ambit of the current theory.

We ran all experiments of this segment using a standard desktop with a GeForce RTX 2060 graphics card. We built custom implementations of $\delta$-Regularized-GClip and standard GClip and, importantly, used the standard Pytorch optimizers for SGD and Adam, which we recall is highly optimized. Hence, we would be

---

[1] https://github.com/mingfeisun/delta-gclip

demonstrating performance of our modification in competitions which are a priori skewed in favour of the existing benchmarks.

In the legends of the figures, the notation 'SGD (0.1)' stands for stochastic gradient descent with learning rate $\eta = 0.1$, notation '$\delta-$GClip $(1; 1; 1e-8)$' stands for $\delta$-Regularized-GClip with $\eta = 1, \gamma = 1, \delta = 1e-8$, and 'GClip $(5; 1)$' for standard gradient clipping with $\eta = 5$, $\gamma = 1$. The notation 'Adam (1)' stands for Adam with $\eta = 1$, and similarly for other hyperparameter choices.

**ResNet-18 on CIFAR-10.** The ResNet-18 was trained using the full training set using mini-batches of size 512. We tested all the following hyperparameter combinations: $\eta \in \{0.0001, 0.001, 0.01, 0.1, 1, 5\}$, $\gamma \in \{0.25, 1, 5, 10\}$ and $\delta \in \{1e-3, 1e-8\}$ for each optimizer. For Adam, only the learning rate ($\eta$) was modified, the rest were left at the PyTorch defaults ($\beta_1 = 0.9$, $\beta_2 = 0.999$, $\varepsilon = 1e-8$). In the case with scheduling the $\eta$ value quoted in the legend denotes the $\eta$ value at epoch 0, i.e. before any reductions by the scheduling algorithm are done.

**Experiments Without Learning Rate Scheduling.** In Figure 2 we only plot the best-performing (in terms of test accuracy) hyperparameter selection for each algorithm.

Figure 2: $\delta$-Regularized-GClip ($\delta$-GClip) is competitive against SOTA heuristics for training ResNet-18 on CIFAR-10 without learning-rate scheduling.

**Experiments With Learning Rate Scheduling.** In Figure 3 we show a repeat of the above experiments and again plot the best-performing hyperparameters. In here, we start at larger $\eta$ values and divide $\eta$ by 10 at epochs 100 and 150, following the setup from Zhang et al. (2020a).

For completeness, in Figure 4 we present a version of the experiment above but without weight-decay for any of the algorithms considered.

We note that the performance of the gradient clipping based algorithms, as well as Adam, do not show significant changes with the removal of weight decay; however, SGD performs significantly worse.

We draw two primary conclusions from these results.

*Firstly,* that a very small value of $\delta$ in $\delta$-Regularized-GClip does not seem to have a significant effect either for loss minimization or test accuracy. The results for $\delta$-Regularized-GClip and standard GClip, set to similar $\eta$

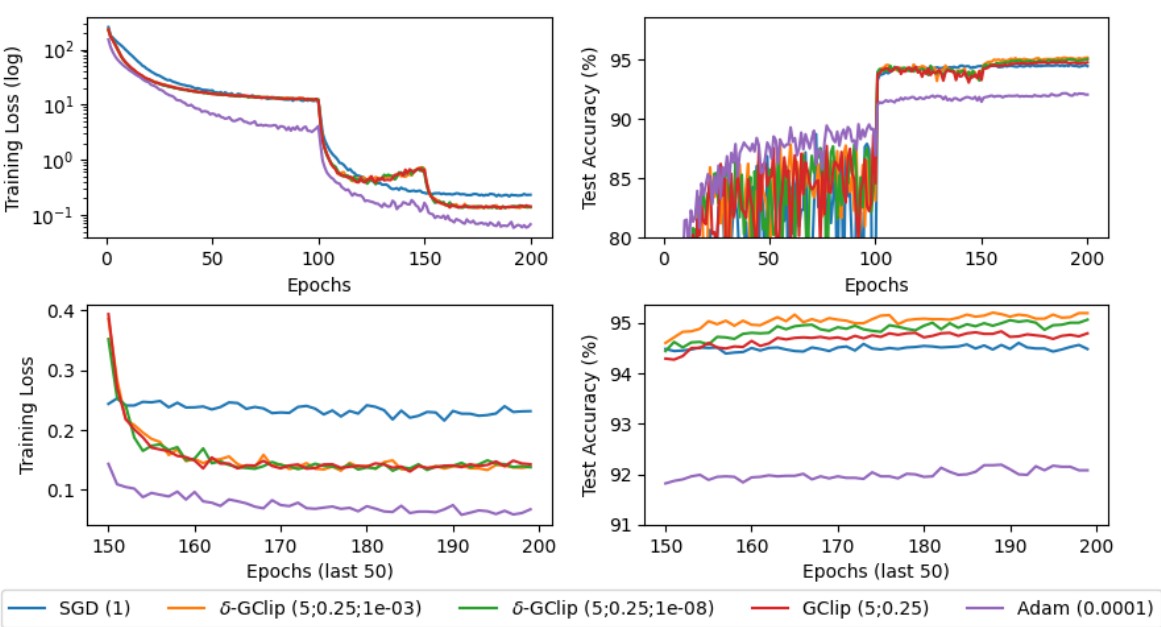

Figure 3: $\delta$-Regularized-GClip ($\delta$-GClip) outperforms other optimizers for training ResNet-18 on CIFAR-10 with learning-rate scheduling.

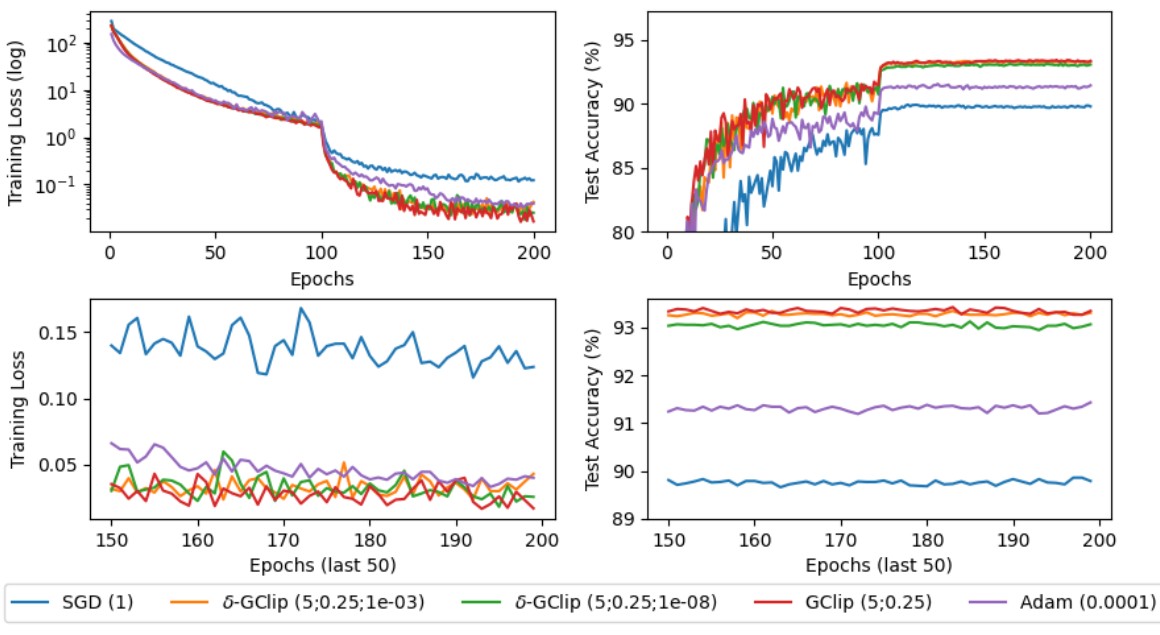

Figure 4: $\delta$-Regularized-GClip ($\delta$-GCLip) matches the best heuristics for training a ResNet-18 on CIFAR-10 with learning-rate scheduling, but no weight-decay.

and $\gamma$ values, are practically identical in either scenarios for all small enough values of $\delta$ tried. As alluded to

in the previous sections, the gradient norm would have to be larger than $\eta\gamma/\delta$ for the lower bound on $h(\boldsymbol{w}_t)$ to be attained, and even for the larger setting of $\delta = 1e-3$ and a typical $\gamma = 0.25$ setting requires a gradient norm of over 250, which is only infrequently seen along the optimization trajectory.

*Secondly,* though Adam attained the best test accuracy without learning rate scheduling by a margin of about ~ 1 percentage point compared to both $\delta$-Regularized-GClip and standard gradient clipping, but all other optimizers surpassed it by ~ 3 percentage points when learning rate scheduling was used. *The best performance with scheduling (which is by our regularized gradient clipping) is better than for any algorithm (ours or not) without scheduling.* Interestingly, with learning rate scheduling Adam performed the best in terms of minimizing the training loss while SGD performed the worst, even though SGD's solution seems to generalize significantly better (as shown by the ~ 3 percentage point higher test accuracy).

The significant ability of $\delta$-regularized gradient clipping to exploit learning rate scheduling motivates an interesting direction for future exploration in theory.

**VAE on Fashion-MNIST.** We performed the VAE training experiment both with and without scheduling when training on the Fashion-MNIST dataset. We tested the following hyperparmeter grid choices: $\eta \in \{1e-5, 1e-4, 1e-3, 1e-2\}$, $\gamma \in \{10, 50, 200, 500\}$, $\delta \in \{0.01, 0.1, 1\}$. We utilize the same scheduling as in the ResNet experiment ($\eta$ division by 10 at epochs 100 and 150), and the results are given in Figure 5.

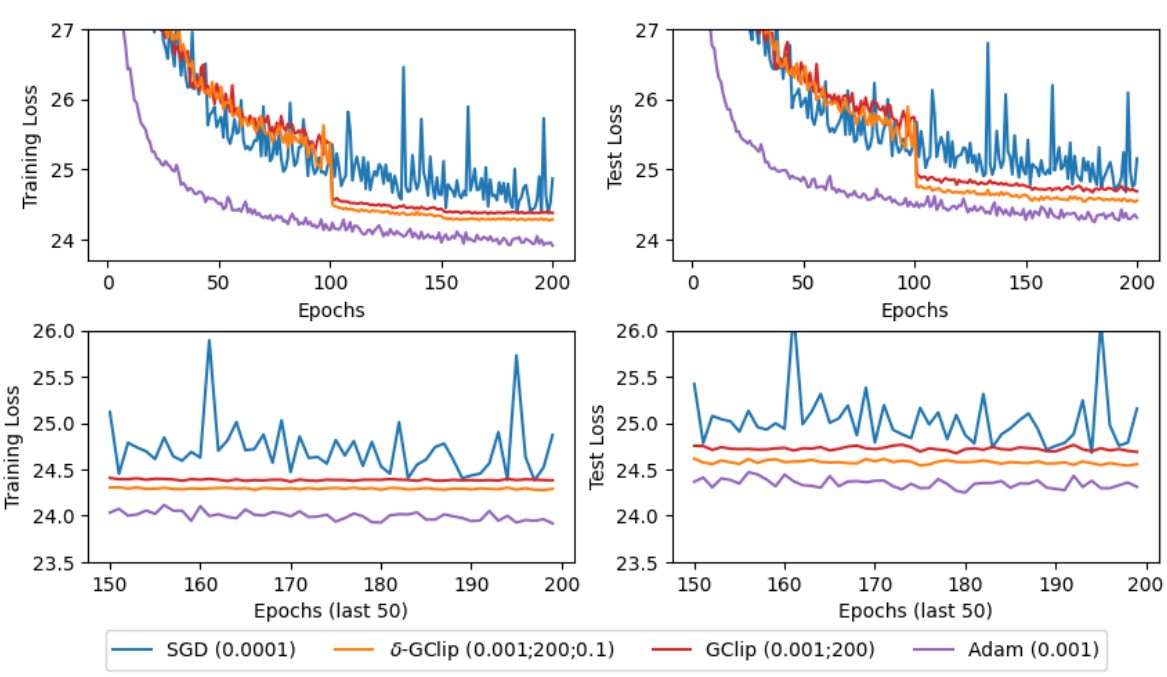

Figure 5: $\delta$-Regularized-GClip ($\delta$-GClip) is competitive against SOTA heuristics for training a VAE on the Fashion-MNIST dataset with learning-rate scheduling.

The VAE results with (and without, though not shown here) learning rate scheduling supports our earlier observations that the added regularization term of $\delta$ helps the performance w.r.t that of GClip at the same values of step-length and clipping threshold, which anyway outperforms SGD. And it is only mildly underperforming with respect to Adam.

We therefore conclude from our experiments that $\delta$-Regularized-GClip clipping remains competitive with current optimizers, while offering the significant benefit of provable deep neural network training.

### 3.2   Evidence for The Performance of $\delta$-Regularized-GClip on Transformers

In this section we focus on benchmark state-of-the-art transformer-based architectures, that are not known to satisfy the $\mu$-PL$^*$ condition — as in the examples of the previous section. Yet, we show that our $\delta$-GClip proposal continues to be competitive against the widely used heuristic of Adam.

In Figure 6 we show that starting from random initialization Adam's ability to train a Vision Transformer (Dosovitskiy et al., 2020) for the classification task on the CIFAR-10 dataset is matched in train and test loss and accuracy by our theoretically grounded proposal of $\delta$-GClip at $\eta = 0.2, \gamma = 1$ and $\delta = 10^{-6}$.

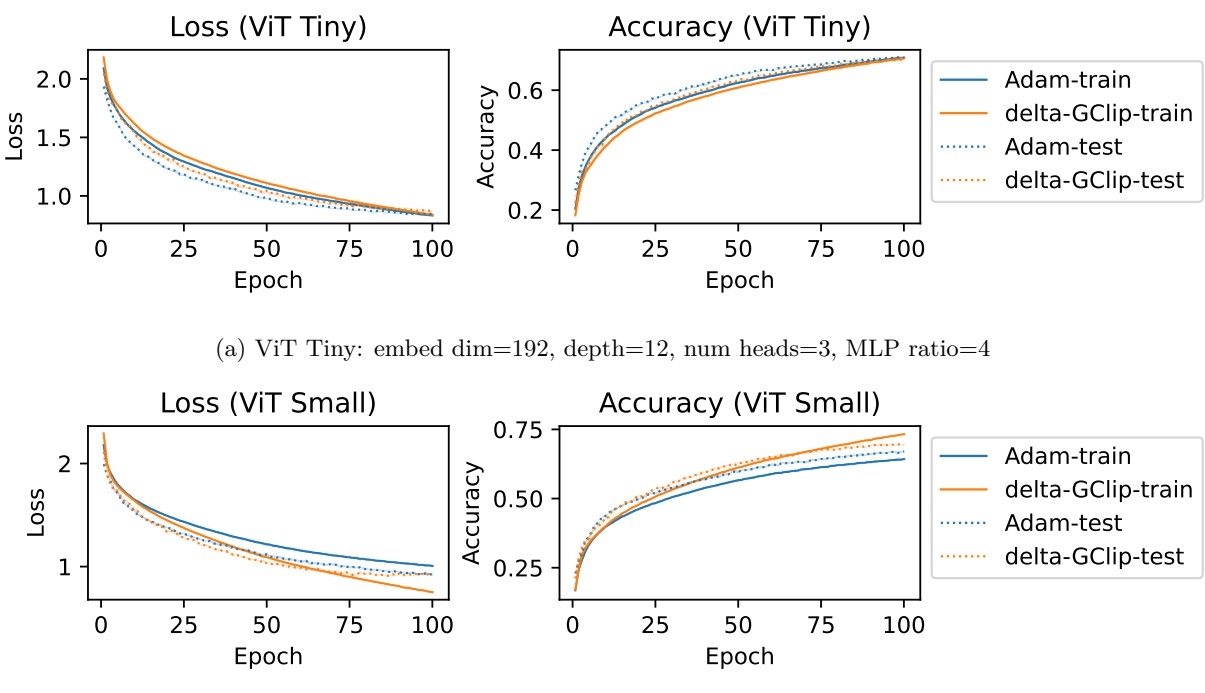

(a) ViT Tiny: embed dim=192, depth=12, num heads=3, MLP ratio=4

(b) ViT Small: embed dim=384, depth=12, num heads=6, MLP ratio=4

Figure 6: $\delta$-GClip can be seen to be competitive against the SOTA heuristic of using Adam for training a Vision Transformer on the CIFAR-10 dataset.

Venturing further, we consider a fine-tuning task on a variant of the BERT (Devlin et al., 2019) model DistilBertForSequenceClassification[2] on the dataset of sst2[3].

On this benchmark we do a hyperparameter search over $\eta$ and $\gamma$ values for $\delta$-GClip to decide on the setting of $\eta = 0.1, \gamma = 0.1$ and as before we set $\delta = 10^{-6}$. As shown in Figure 7, we can see that on the crucial performance metric of test accuracy, yet again $\delta$-GClip matches the standard heuristic which is a mix of standard gradient clipping and Adam. *Hence in this experiment we see that our proposed $\delta$-GClip can compete the blend of adaptive gradient algorithms that is usually deployed to train a BERT.*

In Table 1 we present further performance comparison between Adam and $\delta$-GClip at different $\delta$s are provided, for training the same ViTs as studied in Figure 6. Similar comparisons are presented in Table 2 for the same BERT variant studied in Figure 7. We note that in all these studies the performance metrics reported give the averaged results with the respective standard deviations over multiple runs of the experiment. Thus, from these tables we can robustly conclude that $\delta$-GClip is competitive against Adam in transformer models where similar convergence guarantees are not yet available for $\delta$-GClip, as was presented earlier. Lastly, from these tables we also note that the performance of $\delta$-GClip, is extremely stable with orders of magnitude of variation in $\delta$, in the ranges we consider. Thus the experiments lend themselves to

---

[2]https://huggingface.co/transformers/v3.0.2/model_doc/distilbert.html#distilbertforsequenceclassification
[3]https://huggingface.co/datasets/stanfordnlp/sst2

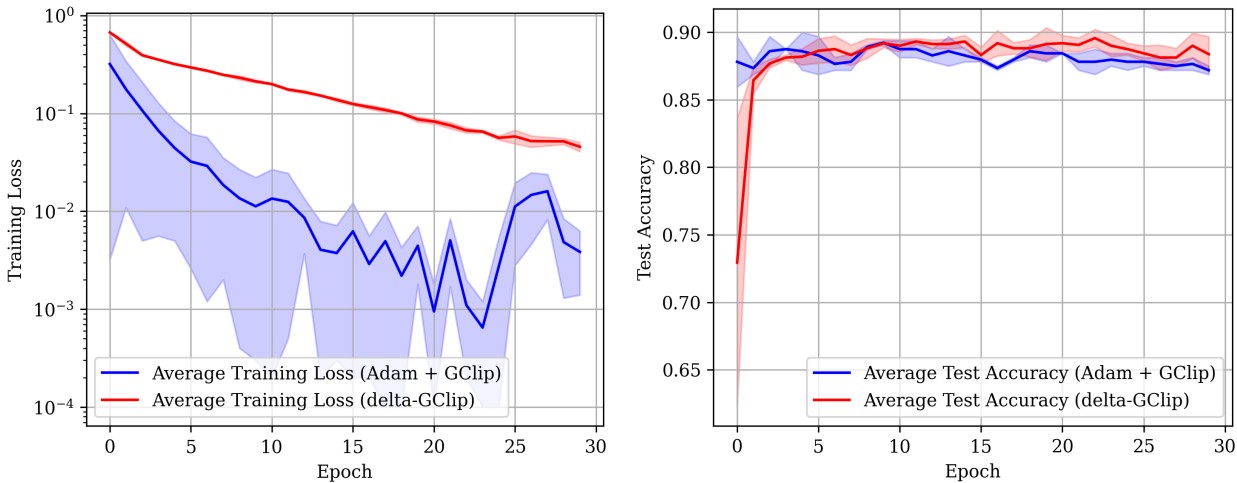

Figure 7: $\delta$-GClip ($\delta = 10^{-6}$) can be seen to be competitive against the SOTA heuristic of using a combination of Adam and GClip for training a variant of the BERT model. (The shaded area shows the variation about mean performance over multiple runs.)

| Transformer Experiments on the CIFAR-10 Dataset | | | |
|---|---|---|---|
| Model | Algorithm | Test Accuracy (avg.) at Last Iterate | Train Error (avg.) at Last Iterate |
| Vision Transformer Tiny embed dim=192, depth=12, num heads=3, MLP ratio=4 | ($\delta = 0.01$) ($\delta = 0.001$) ($\delta = 0.0001$) ($\delta = 0.00001$) (Adam) | $0.697^{\pm0.002}$ $0.701^{\pm0.001}$ $0.701^{\pm0.003}$ $0.694^{\pm0.004}$ $0.708^{\pm0.003}$ | $0.700^{\pm0.003}$ $0.705^{\pm0.002}$ $0.706^{\pm0.002}$ $0.703^{\pm0.002}$ $0.848^{\pm0.005}$ |
| Vision Transformer Small embed dim=384, depth=12, num heads=6, MLP ratio=4 | ($\delta = 0.01$) ($\delta = 0.001$) ($\delta = 0.0001$) ($\delta = 0.00001$) (Adam) | $0.751^{\pm0.003}$ $0.761^{\pm0.005}$ $0.762^{\pm0.009}$ $0.754^{\pm0.005}$ $0.672^{\pm0.004}$ | $0.697^{\pm0.001}$ $0.693^{\pm0.003}$ $0.695^{\pm0.003}$ $0.694^{\pm0.003}$ $0.975^{\pm0.003}$ |

Table 1: Comparison of Performance between $\delta$-Regularized-GClip (at different $\delta$) and Adam for experiments on Vision Transformers.

the interpretation that the introduction of the small $\delta$ parameter in standard GClip is a theoretical tool that "stabilizes" standard clipping in ways that lend it to provable guarantees without making any material difference to its performance.

| Transformer Experiments on the sst2 Classification Dataset | | | |
|---|---|---|---|
| Model | Algorithm | Test Accuracy (avg.) at Last Iterate | Train Error (avg.) at Last Iterate |
| DistillBertforSequenceClassification | ($\delta = 8 \times 10^{-6}$) ($\delta = 10^{-6}$) ($\delta = 1.25 \times 10^{-7}$) (Adam) | $0.879^{\pm0.018}$ $0.879^{\pm0.018}$ $0.879^{\pm0.018}$ $0.871^{\pm0.004}$ | $0.045^{\pm0.003}$ $0.045^{\pm0.003}$ $0.045^{\pm0.003}$ $0.003^{\pm0.003}$ |

Table 2: Comparison of Performance between $\delta$-Regularized-GClip (at different $\delta$) and Adam for experiments on a BERT variant.

## 4 Related Works

In this section, we will summarize the state-of-the-art literature on standard clipping and provable deep learning optimization algorithms that are adaptive. At the very outset, we note that the Neural Tangent Kernel (NTK) approach to provable deep learning at large widths Jacot et al. (2018); Du et al. (2018); Chizat et al. (2019); Allen-Zhu et al. (2019b;a); Zou et al. (2020) is not within the scope of our review since, in general, they do not use step-length scheduling. Furthermore, it is also known that NTK based predictors are outperformed by standard deep learning architectures (Chen et al., 2020b; Arora et al., 2019).

**Literature Review of Theory for Adam.** Adam was proposed in Kingma & Ba (2014) as an adaptive algorithm which requires hyperparameters $\beta_1, \beta_2 \in [0, 1)$ to control the decay rates of exponential moving average estimates for the gradients and the squared gradients, respectively. In Reddi et al. (2018) it was proved that, for common hyperparameter choices $(\beta_1 < \sqrt{\beta_2})$, there exists a stochastic convex optimization problem where Adam does not converge. They presented a modification to Adam that provably converges for online convex optimization. In De et al. (2018) for the first time the convergence of Adam was established in the deterministic case, without the use of convexity, but leveraging Lipschitz smoothness and a bounded gradient norm.

For the same optimization target as above, in Chen et al. (2019), a convergence rate of $O(\log T / \sqrt{T})$ was shown for Adam-like adaptive gradient algorithms under the assumption of a bounded gradient oracle. Later, a burn-in stage was added in Staib et al. (2019) to prove a $O(1/\sqrt{T})$ convergence rate. In Chen et al. (2020a), a partial adaptive parameter was introduced, and they proved convergence to criticality for a class of adaptive gradient algorithms, which does not include RMSProp. It was shown in Zou et al. (2019) that generic Adam (including RMSProp) converges with high probability under certain decaying conditions on $\beta_2$ and step size, in contrast to the usual implementations. In Ward et al. (2019), similar convergence results were proved for AdaGrad, which is a special case of RMSProp.

**Review of Theory for Adaptive Gradient Methods Training Neural Nets.** In contrast to the convergence to criticality results mentioned above, there have also been works providing guarantees of convergence to a global minima for adaptive methods in shallow neural net training scenarios. Wu et al. (2019) provides a proof of the convergence of the AdaLoss adaptive algorithm to global minima on two-layer neural nets, under widths large enough to be in the NTK regime. Zou et al. (2023) provides a proof of Adam's global convergence on two-layer convolutional neural networks to a zero-error solution whilst utilising weight decay regularization. Further, Zou et al. (2023) provide evidence that although both GD and Adam converge to zero-error solutions, GD's solution generalises significantly better. In the context of Generative Adversarial Networks (GANs), Dou & Li (2021) analysed the performance of Adam-like algorithms and proved the convergence of Extra Gradient AMSGrad to an $\varepsilon$-stationary point under novel assumptions they motivated.

**Literature Review of Gradient Clipping.** In the smooth non-convex case, Zhang et al. (2020a) proved the convergence of deterministic gradient clipping to an $\varepsilon$-stationary point under a new smoothness notion that is strictly weaker than standard Lipschitz smoothness. Their provided iteration complexity implies that gradient clipping can converge faster than gradient descent (in constants), while achieving $\mathcal{O}(\varepsilon)$-criticality in $\mathcal{O}(\varepsilon^{-2})$ steps. They provide a similar analysis in the stochastic case, with the additional assumption of either a bound on the noise of the stochastic gradient or its distribution being symmetric sub-Gaussian. It is important to note that the provided stochastic iteration complexity does not supersede that of SGD in the general case. They had pointed out, possibly for the first time, that gradient clipping can converge, in deterministic as well as noisy settings, on smooth functions without the need for gradients to be bounded.

In Zhang et al. (2020b), Lipschitz smoothness is utilized while working with non-convex otimization targets and heavy-tailed gradient stochasticity to achieve $\mathcal{O}(1/t^{\frac{1}{4}})$ close convergence to criticality in $t$ steps, which matches that of SGD in the non-heavy-tailed setting. The same work gave a lower bound in the same setting, which matches up to constants the run-time given above and thus proving that their convergence rate is worst-case optimal. Furthermore, the said work also considered non-smooth but strongly convex functions with a bound on the expected norm of the stochastic gradients—which we recall had appeared earlier in Shamir & Zhang (2013) for non-heavy tailed settings—and achieve the same convergence, implying that the convergence rate is optimal even in the Lipschitz smooth and strongly convex setting.

We posit that from above kinds of analysis of adaptive algorithms (including GClip), either for depth-2 neural networks or in the more general (non-)convex settings, there is no obvious path towards provable convergence guarantees in deep neural network training for adaptive gradient algorithms. However, in the recent work of Liu et al. (2022), convergence guarantees were proven for (S)GD for sufficiently wide, and arbitrarily deep neural networks, by leveraging the novel PL* condition that the same paper proved to be true for squared losses for such neural nets. Next we will briefly review those results.

**Review of the** PL* **Condition.**    Our motivation for studying the convergence characteristics of algorithms under the PL* condition comes from Liu et al. (2022), where it is proved that overparametrized feedforward, convolutional and residual (ResNet) neural networks can all satisfy the PL* condition within a finite radius of the initialization, if they are sufficiently wide. In particular, Liu et al. (2022) showed the following.

**Theorem 4.1.** Consider any neural network of the form described in Definition 4. If weight matrices are randomly initialized s.t. $W_0^{(l)} \sim \mathcal{N}(0, I_{m_l \times m_{l-1}})$ for $l \in [0, L+1]$, and defining $\lambda_0 := \lambda_{\min}(K_{\mathcal{F}}(\boldsymbol{w}_0)) > 0$ where $K_{\mathcal{F}}(\boldsymbol{w}) = \mathcal{D}\mathcal{F}(\boldsymbol{w})\mathcal{D}\mathcal{F}(\boldsymbol{w})^\top$, then for any $\mu \in (0, \lambda_0 \rho^2)$ and the minimum layer width of the network being

$$m = \tilde{\Omega}\left(\frac{nR^{6L+2}}{(\lambda_0 - \mu\rho^{-2})^2}\right), \tag{5}$$

the $\mu$-PL* condition holds for the square loss in the ball $\mathrm{B}(\boldsymbol{w}_0, R)$ where $R$ is a finite radius.

Therefore, a path opened up, that by proving that the iterates of our $\delta$-Regularized-GClip algorithm never leave a ball of finite radius, and proving the convergence of $\delta$-Regularized-GClip on locally smooth $\mu$-PL* functions, we can argue for the algorithm's convergence to the loss global minima in such neural nets.

## 5   Proof of the Main Theory Result for $\delta$-**Regularized-GClip**

Towards giving the proof of the main theorem for $\delta$-GClip algorithm (Theorem 2.1), we begin with listing some lemmas that are needed for the proof.

### 5.1   Preparatory Lemmas

**Lemma 5.1.** Corresponding to constants $a, b > 0$ and $a\mu < 1$ suppose a loss function $\mathcal{L}$ is $\beta$-smooth, $\min \mathcal{L} = 0$, and satisfies the $\mu$-PL* condition within a Euclidean ball $\mathrm{B}(\boldsymbol{w}_0, \mathrm{R})$, with $R \geq \dfrac{b\sqrt{2\beta}\sqrt{\mathcal{L}(\boldsymbol{w}_0)}}{1 - \sqrt{1 - \frac{1}{2} \cdot a\mu}}$. Then there exists a global minimizer $\boldsymbol{w}_* \in B(\boldsymbol{w}_0, R)$ of $\mathcal{L}$, such that $\mathcal{L}(\boldsymbol{w}_*) = 0$. Furthermore, given a first-order adaptive step size algorithm of the form

$$\boldsymbol{w}_{t+1} = \boldsymbol{w}_t - h(\boldsymbol{w}_t) \cdot \nabla \mathcal{L}(\boldsymbol{w}_t), \tag{6}$$

where $h(\boldsymbol{w}_t)$ is a time/iterate-dependent function such that $0 < a \leq h(\boldsymbol{w}_t) \leq b < \min\{\frac{1}{\beta}, \frac{1}{\mu}\}$, then the algorithm will converge with rate

$$\mathcal{L}(\boldsymbol{w}_t) \leq \mathcal{L}(\boldsymbol{w}_0)(1 - \frac{1}{2} \cdot a\mu)^t. \tag{7}$$

**Lemma 5.2.** The $\delta$-Regularized-GClip step size $h(\boldsymbol{w})$ is bounded $\eta\delta \leq h(\boldsymbol{w}) \leq \eta$, given that $0 < \delta < 1$.

**Lemma 5.3.** ($\delta$-Regularized-GClip Converges on smooth PL* functions.) Corresponding to positive constants $\eta, \delta, \beta, \mu$ such that $\eta < \min\{1/\beta, 1/\mu\}$ and $0 < \delta < 1$, suppose there exists a loss function $\mathcal{L}$ that is $\beta$-smooth, lower-bounded by 0, and satisfies the $\mu$-PL* condition within an Euclidean ball $\mathrm{B}(\boldsymbol{w}_0, R)$ where $R \geq \frac{\eta\sqrt{2\beta}\sqrt{\mathcal{L}(\boldsymbol{w}_0)}}{1 - \sqrt{1 - \frac{1}{2} \cdot \eta\delta\mu}}$. Then there exists a global minimizer $\boldsymbol{w}_* \in B(\boldsymbol{w}_0, R)$ of $\mathcal{L}$, such that $\mathcal{L}(\boldsymbol{w}_*) = 0$. Furthermore, $\delta$-Regularized-GClip will converge at rate

$$\mathcal{L}(\boldsymbol{w}_t) \leq \mathcal{L}(\boldsymbol{w}_0)(1 - \frac{1}{2} \cdot \eta\delta\mu)^t. \tag{8}$$

The proofs for Lemmas 5.1, 5.2 and 5.3 can be found in Subsection 5.3.

## 5.2 Proof of Theorem 2.1

*Proof.* Firstly, we invoke the assumption that the initialization is s.t that the conditions of Theorem 4.1 apply, which we know from therein to be a high-probability event. In particular we conclude that $\mathcal{L}$ satisfies $\mu$-PL* within a finite ball $B(\boldsymbol{w}_0, R)$ for some $R > 0$ and that the tangent kernel at initialization is positive definite.

If $L_\sigma$ and $\beta_\sigma$ are the Lipschitz constant and the Lipschitz smoothness coefficients for the activation $\sigma$, then it was shown in Liu et al. (2022) that for the prediction map $\mathcal{F}$ we have its Lipschitz constant,

$$L_\mathcal{F} \leq L_\sigma \left( \sqrt{\|K_\mathcal{F}(\boldsymbol{w}_0)\|} + R\sqrt{n} \cdot O(R^{3L}/\sqrt{m}) \right),$$

as well as its smoothness constant,

$$\beta_\mathcal{F} \leq \beta_\sigma L_\sigma \left( \sqrt{\|K_\mathcal{F}(\boldsymbol{w}_0)\|} + R\sqrt{n} \cdot O(R^{3L}/\sqrt{m}) \right) + L_\sigma \cdot O(R^{3L}/\sqrt{m}),$$

where $K_\mathcal{F}$ is the neural tangent kernel (recall that $K_\mathcal{F} = \mathcal{D}\mathcal{F}(\boldsymbol{w})\mathcal{D}\mathcal{F}(\boldsymbol{w})^\top$).

By plugging in the lower bound on $m$ specified in the theorem, we get that both $L_\mathcal{F}$ and $\beta_\mathcal{F}$ are upper bounded by a constant and thus $m$-independent. If $H_\mathcal{L}$ is the Hessian of the loss function, then by Liu et al. (2022) we also have that

$$\beta_\mathcal{L} = \sup_{\boldsymbol{w} \in B(\boldsymbol{w}_0, R)} \|H_\mathcal{L}(\boldsymbol{w})\| \leq L_\mathcal{F}^2 + \beta_\mathcal{F} \cdot \|\mathcal{F}(\boldsymbol{w}_0) - \boldsymbol{y}\|.$$

By Jacot et al. (2018), we have that $\|\mathcal{F}(\boldsymbol{w}_0) - \boldsymbol{y}\|$ is also $m$-independent with high probability for the given size of the net. Therefore, $\mathcal{L}$ can be said to be $\beta_\mathcal{L}$-smooth within $B(\boldsymbol{w}_0, R)$, where $\beta_\mathcal{L}$ is $m$-independent and thus $R$-independent. Hence, we can say that for every $R > 0$, for some width which satisfies the given condition, the loss function is $\beta$-smooth (and by Theorem 4.1, $\mu$-PL*) in $B(\boldsymbol{w}_0, R)$ with high probability.

Thus far the argument above was parametric in $R$. But given that we satisfy all the conditions to invoke Lemma 5.3 we can compute from it the minimum $R$ value required such that the iterates of regularized gradient clipping never leave $B(\boldsymbol{w}_0, R)$, i.e

$$R = \frac{\eta\sqrt{2\beta}\sqrt{\mathcal{L}(\boldsymbol{w}_0)}}{1 - \sqrt{1 - \frac{1}{2} \cdot \eta\delta\mu}},$$

and conclude that $\delta$-Regularized-GClip converges to a zero-loss solution within $B(\boldsymbol{w}_0, R)$ at a convergence rate of $\mathcal{L}(\boldsymbol{w}_t) \leq \mathcal{L}(\boldsymbol{w}_0)(1 - \eta\delta\mu)^t$. $\qquad\square$

## 5.3 Proofs of the Lemmas

*Proof.* **(of Lemma 5.1)** We shall prove the theorem by induction and our hypothesis is that, up to step $t$, $\boldsymbol{w}_t \in B(\boldsymbol{w}_0, R)$ for the given $R$, $\mathcal{L}(\boldsymbol{w}_t) \leq \mathcal{L}(\boldsymbol{w}_0)(1 - \frac{1}{2} \cdot a\mu)^t$ and thus up to $t$ the algorithm explored a region where the $\mu$-PL* condition holds. The base case is trivial, when $t = 0$ then $\boldsymbol{w}_0 \in B(\boldsymbol{w}_0, R)$. Now we set out to prove that these continue to hold at $t + 1$ too.

From the assumptions that, $\mathcal{L}$ is $\beta$-smooth, we have

$$\mathcal{L}(\boldsymbol{w}_{t+1}) - \mathcal{L}(\boldsymbol{w}_t) - \nabla\mathcal{L}(\boldsymbol{w}_t)^\top(\boldsymbol{w}_{t+1} - \boldsymbol{w}_t) \leq \frac{\beta}{2}\|\boldsymbol{w}_{t+1} - \boldsymbol{w}_t\|^2. \tag{9}$$

As $h(\boldsymbol{w}_t) < \min\{\frac{1}{\beta}, \frac{1}{\mu}\}$, we have that $\frac{1}{h(\boldsymbol{w}_t)} > \beta$, hence we relax the above inequality to get, $\mathcal{L}(\boldsymbol{w}_{t+1}) - \mathcal{L}(\boldsymbol{w}_t) - \nabla\mathcal{L}(\boldsymbol{w}_t)^\top(\boldsymbol{w}_{t+1} - \boldsymbol{w}_t) \leq \frac{1}{2h(\boldsymbol{w}_t)}\|\boldsymbol{w}_{t+1} - \boldsymbol{w}_t\|^2$.

Using the definition of the algorithm, that $\boldsymbol{w}_{t+1} - \boldsymbol{w}_t = -h(\boldsymbol{w}_t)\nabla\mathcal{L}(\boldsymbol{w}_t)$, we can rearrange the above to get

$$\mathcal{L}(\boldsymbol{w}_{t+1}) - \mathcal{L}(\boldsymbol{w}_t) \leq -\frac{h(\boldsymbol{w}_t)}{2}\|\nabla\mathcal{L}(\boldsymbol{w}_t)\|^2. \tag{10}$$

Next, we use the induction hypothesis for the $\mu$-PL$^*$ condition at the current iterate, $\|\nabla\mathcal{L}(\boldsymbol{w}_t)\|^2 \geq \mu\mathcal{L}(\boldsymbol{w}_t)$, to get

$$\mathcal{L}(\boldsymbol{w}_{t+1}) - \mathcal{L}(\boldsymbol{w}_t) \leq -\frac{h(\boldsymbol{w}_t)}{2}\|\nabla\mathcal{L}(\boldsymbol{w}_t)\|^2 \leq -\frac{h(\boldsymbol{w}_t)\mu}{2}\mathcal{L}(\boldsymbol{w}_t).$$

And the above can be rearranged to

$$\mathcal{L}(\boldsymbol{w}_{t+1}) \leq (1 - \frac{1}{2}\cdot h(\boldsymbol{w}_t)\mu)\mathcal{L}(\boldsymbol{w}_t). \tag{11}$$

Note that for the convergence rate to hold, $h(\boldsymbol{w}_t)$ must be bounded such that $\forall t$, $(1 - \frac{1}{2}\cdot h(\boldsymbol{w}_t)\mu) \in (0,1)$ and this follows from the bounds on $a, b$. We then unroll the recursion to get

$$\begin{aligned}\mathcal{L}(\boldsymbol{w}_{t+1}) &\leq \mathcal{L}(\boldsymbol{w}_0)\cdot\prod_{i=0}^{t}(1 - \frac{h(\boldsymbol{w}_i)\mu}{2}) \\ &\leq \mathcal{L}(\boldsymbol{w}_0)(1 - \frac{1}{2}\cdot a\mu)^{t+1}\end{aligned} \tag{12}$$

where the last inequality comes from $0 < a \leq h(\boldsymbol{w}_t)$. Therefore, assuming that the convergence rate holds till time $t$ implies that it also holds till $t+1$.

Next we embark on proving that $\boldsymbol{w}_{t+1} \in B(\boldsymbol{w}_0, R)$. From the algorithm's update equation, the triangle inequality, and recalling that $h(\boldsymbol{w}_t) \leq b$, we get

$$\|\boldsymbol{w}_{t+1} - \boldsymbol{w}_0\| \leq \sum_{i=0}^{t}\|h(\boldsymbol{w}_i)\cdot\nabla\mathcal{L}(\boldsymbol{w}_i)\| \leq b\sum_{i=0}^{t}\|\nabla\mathcal{L}(\boldsymbol{w}_i)\|. \tag{13}$$

We can rearrange the $\beta$-smoothness inequality from equation (9) and apply Cauchy-Schwarz, to get

$$0 \leq \frac{\beta}{2}\|\boldsymbol{w}_{t+1} - \boldsymbol{w}_t\|^2 + \|\nabla\mathcal{L}(\boldsymbol{w}_t)\|\|\boldsymbol{w}_{t+1} - \boldsymbol{w}_t\| + \mathcal{L}(\boldsymbol{w}_t) - \mathcal{L}(\boldsymbol{w}_{t+1}). \tag{14}$$

We can relax the above inequality by dropping the $\mathcal{L}(\boldsymbol{w}_{t+1})$ term and treating the above as a quadratic in $\|\boldsymbol{w}_{t+1} - \boldsymbol{w}_t\|$, to then conclude that the inequality only holds if the discriminant is non-positive, $\|\nabla\mathcal{L}(\boldsymbol{w}_t)\| \leq \sqrt{2\beta\mathcal{L}(\boldsymbol{w}_t)}$. Substituting this into equation (13) we get

$$\|\boldsymbol{w}_{t+1} - \boldsymbol{w}_0\| \leq b\sum_{i=0}^{t}\sqrt{2\beta\mathcal{L}(\boldsymbol{w}_i)}. \tag{15}$$

Using the assumed convergence rate till the current iterate we get

$$\|\boldsymbol{w}_{t+1} - \boldsymbol{w}_0\| \leq b\sqrt{2\beta}\sqrt{\mathcal{L}(\boldsymbol{w}_0)}\cdot\left(\sum_{i=0}^{t}\prod_{j=0}^{i}(1 - \frac{1}{2}\cdot h(\boldsymbol{w}_j)\mu)^{1/2}\right). \tag{16}$$

Since $a \leq h(\boldsymbol{w}_t) < 1/\mu$, we have, $0 < 1 - \frac{1}{2}\cdot h(\boldsymbol{w}_t)\mu < 1 - \frac{1}{2}\cdot a\mu < 1$. Thus we get

$$\|\boldsymbol{w}_{t+1} - \boldsymbol{w}_0\| \leq b\sqrt{2\beta}\sqrt{\mathcal{L}(\boldsymbol{w}_0)}\cdot\left(\sum_{i=0}^{t}(1 - \frac{1}{2}\cdot a\mu)^{i/2}\right). \tag{17}$$

Upper-bounding the above by the closed-form expression for the infinite geometric series, we get

$$\|\boldsymbol{w}_{t+1} - \boldsymbol{w}_0\| \leq \frac{b\sqrt{2\beta}\sqrt{\mathcal{L}(\boldsymbol{w}_0)}}{1 - \sqrt{1 - \frac{1}{2}\cdot a\mu}} \leq R. \tag{18}$$

The last inequality follows by the definition of $R$ and hence we have proven that $\boldsymbol{w}_{t+1} \in B(\boldsymbol{w}_0, R)$, and hence up to time $t+1$ the algorithm is still exploring the region within which the $\mu$-PL$^*$ condition holds.

Thus induction follows and we have that $\forall t$, $\boldsymbol{w}_t \in B(\boldsymbol{w}_0, R)$ and $\mathcal{L}(\boldsymbol{w}_t) \leq \mathcal{L}(\boldsymbol{w}_0)(1 - \frac{1}{2}\cdot a\mu)^t$. $\qquad\square$

**Proof of $\delta$-Regularized-GClip Having a Bounded Step Size**

*Proof.* **(of Lemma 5.2)** Utilising $\delta$-Regularized-GClip's definition for $h$, we get that if $\|\nabla\mathcal{L}(\boldsymbol{w}_t)\| \geq \gamma/\delta$, then $h(\boldsymbol{w}_t) = \min\{\eta, \eta\delta\}$. Otherwise, if $\|\nabla\mathcal{L}(\boldsymbol{w}_t)\| < \gamma/\delta$, then $h(\boldsymbol{w}_t) = \min\{\eta, \ \eta\gamma/\|\nabla\mathcal{L}(\boldsymbol{w}_t)\|\}$. The smallest possible $h$ for the above would be if $\|\nabla\mathcal{L}(\boldsymbol{w}_t)\|$ was as large as it could be, which would result in $h(\boldsymbol{w}_t) = \min\{\eta, \eta\delta\}$. As $\delta < 1$, we conclude $0 < \eta\delta \leq h(\boldsymbol{w}_t) \leq \eta$. $\qquad\square$

**Proof of $\delta$-Regularized-GClip Convergence on Smooth $\mathrm{PL}^*$ Functions**

*Proof.* **(of Lemma 5.3)** From Lemma 5.2, we know that $\delta$-Regularized-GClip satisfies the condition $0 < \eta\delta \leq h(\boldsymbol{w}_t) \leq \eta$. Therefore, by setting $\eta < \min\{\frac{1}{\beta}, \frac{1}{\mu}\}$ and $\delta < 1$, we can apply Lemma 5.1 and obtain the convergence rate

$$\mathcal{L}(\boldsymbol{w}_t) \leq \mathcal{L}(\boldsymbol{w}_0)(1 - \frac{1}{2} \cdot \eta\delta\mu)^t, \tag{19}$$

as well as that the $\mathrm{PL}^*$ condition must hold within a ball $\mathrm{B}(\boldsymbol{w}_0, \mathrm{R})$, where $R \geq \dfrac{\eta\sqrt{2\beta}\sqrt{\mathcal{L}(\boldsymbol{w}_0)}}{1 - \sqrt{1 - \frac{1}{2} \cdot \eta\delta\mu}}$. $\qquad\square$

## 6 Conclusion

In this work, we have presented a new adaptive gradient algorithm, $\delta$-Regularized-GClip, that provably trains deep neural networks (at any depth) with arbitrary data and while training on the squared loss. To the best of our knowledge, such a guarantee does not exist for *any* previously known adaptive gradient method. Additionally, we have also given experimental evidence that our algorithm is competitive with the deep learning algorithms in current use, and sometimes outperforming them. Our proof critically hinges on the interplay between the modification we do to standard gradient clipping and the $\mu$-$\mathrm{PL}^*$ condition that has previously been shown to be true for squared losses on deep neural nets of sufficient width.

Our work suggests an immediate direction of future research into establishing convergence guarantees for regularized gradient clipping on other standard losses in use, such as cross-entropy and for nets with ReLU activation. Further, the demonstrated success of $\delta$-GClip on certain transformer architectures motivates study of the possible validity of the $\mu$-$\mathrm{PL}^*$ condition for these models too.

Lastly, we note that recently reported heuristics which are particularly good for LLM training, cf. Liu et al. (2024), can also be seen as modifications of the clipping algorithm. We envisage exciting lines of investigation that could open up in trying to explore the efficacy of these new developments crossed with the provably performant modification of gradient clipping that we have instantiated here.

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

# A    A Proof of Convergence for Stochastic $\delta$-Regularized-GClip

We start by proving a more general result as follows,

**Theorem A.1.** Given Assumptions 1, 2 and 3, and for an arbitrary choice of $\epsilon > 0$, consider

$$1 > \delta > \frac{\left(1 + \left(\frac{\epsilon}{\theta}\right)^2\right)}{\left(1 + 3\left(\frac{\epsilon}{\theta}\right)^2\right)}$$

and

$$0 < \eta < \frac{\delta\left(1 + 3\left(\frac{\epsilon}{\theta}\right)^2\right) - \left(1 + \left(\frac{\epsilon}{\theta}\right)^2\right)}{2\beta(1 + \left(\frac{\epsilon}{\theta}\right)^2)}.$$

Then, stochastic $\delta$-Regularized-GClip satisfies the following inequality over any $T > 0$ iterations:

$$\min_{t=1,\dots,T} \mathbb{E}\left[\|\nabla\mathcal{L}(\boldsymbol{w}_t)\|^2\right] \le \epsilon^2 + \frac{1}{T} \cdot \frac{\mathcal{L}(\boldsymbol{w}_1)}{\left(\frac{\eta}{2}(3\delta - 1) - \beta\eta^2\right)}. \tag{20}$$

It's clear from above that we can choose any $\epsilon > 0$ howsoever small and $T > 0$ howsoever large and have the minimum value over iterates of the expected gradient norm be similarly small. To prove Theorem A.1 we need the following two lemmas.

**Lemma A.2.**

$$\mathbb{E}\left[h(\boldsymbol{g}_t)^2\langle \boldsymbol{g}_t - \nabla\mathcal{L}(\boldsymbol{w}_t), \nabla\mathcal{L}(\boldsymbol{w}_t)\rangle \mid \boldsymbol{w}_t\right] \le \eta^2\theta\|\nabla\mathcal{L}(\boldsymbol{w}_t)\|. \tag{21}$$

*Proof.* We begin by employing Cauchy-Schwarz and Assumption 1 to get

$$\begin{aligned}
\mathbb{E}\left[h(\boldsymbol{g}_t)^2\langle \boldsymbol{g}_t - \nabla\mathcal{L}(\boldsymbol{w}_t), \nabla\mathcal{L}(\boldsymbol{w}_t)\rangle \mid \boldsymbol{w}_t\right] &\le \mathbb{E}\left[h(\boldsymbol{g}_t)^2\|\boldsymbol{g}_t - \nabla\mathcal{L}(\boldsymbol{w}_t)\| \mid \boldsymbol{w}_t\right]\|\nabla\mathcal{L}(\boldsymbol{w}_t)\| \\
&\le \mathbb{E}\left[h(\boldsymbol{g}_t)^2 \mid \boldsymbol{w}_t\right]\|\nabla\mathcal{L}(\boldsymbol{w}_t)\|\theta \\
&\le \eta^2\theta\|\nabla\mathcal{L}(\boldsymbol{w}_t)\| \tag{22}
\end{aligned}$$

where in the last inequality we invoked the fact that $h(\boldsymbol{g}_t) \le \eta$. $\qquad\square$

**Lemma A.3.**

$$\mathbb{E}\left[(-h(\boldsymbol{g}_t))\langle \boldsymbol{g}_t - \nabla\mathcal{L}(\boldsymbol{w}_t), \nabla\mathcal{L}(\boldsymbol{w}_t)\rangle \mid \boldsymbol{w}_t\right] \le (\eta - \eta\delta) \cdot \theta \cdot \|\nabla\mathcal{L}(\boldsymbol{w}_t)\|.$$

*Proof.* Because $\boldsymbol{g}_t$ is an unbiased gradient estimate we have,

$$\mathbb{E}[(-h(\boldsymbol{g}_t)) \cdot \langle \boldsymbol{g}_t - \nabla\mathcal{L}(\boldsymbol{w}_t), \nabla\mathcal{L}(\boldsymbol{w}_t)\rangle|\boldsymbol{w}_t] = \mathbb{E}[(\eta - h(\boldsymbol{g}_t)) \cdot \langle \boldsymbol{g}_t - \nabla\mathcal{L}(\boldsymbol{w}_t), \nabla\mathcal{L}(\boldsymbol{w}_t)\rangle|\boldsymbol{w}_t]$$

Noting that $0 \le \eta - h(\boldsymbol{g}_t) \le \eta - \eta\delta$, we get

$$\mathbb{E}[(-h(\boldsymbol{g}_t)) \cdot \langle \boldsymbol{g}_t - \nabla\mathcal{L}(\boldsymbol{w}_t), \nabla\mathcal{L}(\boldsymbol{w}_t)\rangle|\boldsymbol{w}_t] \le (\eta - \eta\delta) \cdot \theta \cdot \|\nabla\mathcal{L}(\boldsymbol{w}_t)\|$$

which establishes the claimed inequality. $\qquad\square$

## A.1    Proof of Theorem A.1

*Proof.* We parameterize the line from $\boldsymbol{w}_t$ to $\boldsymbol{w}_{t+1}$ as $\kappa(t) = t\boldsymbol{w}_t + (1 - t)\boldsymbol{w}_{t+1}$ and applying the Taylor's expansion and then Cauchy-Schwarz formula for the loss evaluated at its end-point we get

$$\begin{aligned}
\mathbb{E}[\mathcal{L}(\boldsymbol{w}_{t+1}) \mid \boldsymbol{w}_t] &\le \mathbb{E}\Bigg[\mathcal{L}(\boldsymbol{w}_t) - h(\boldsymbol{g}_t)\langle \boldsymbol{g}_t, \nabla\mathcal{L}(\boldsymbol{w}_t)\rangle \\
&\qquad + \frac{1}{2}\int_0^1 (\boldsymbol{w}_{t+1} - \boldsymbol{w}_t)^\top \nabla^2\mathcal{L}(\kappa(s))(\boldsymbol{w}_{t+1} - \boldsymbol{w}_t)\,ds \mid \boldsymbol{w}_t\Bigg] \\
&\le \mathcal{L}(\boldsymbol{w}_t) - \mathbb{E}[h(\boldsymbol{g}_t)\langle \boldsymbol{g}_t, \nabla\mathcal{L}(\boldsymbol{w}_t)\rangle \mid \boldsymbol{w}_t] \\
&\qquad + \frac{\mathbb{E}[\|\boldsymbol{w}_{t+1} - \boldsymbol{w}_t\|^2 \mid \boldsymbol{w}_t]}{2}\int_0^1 \left\|\nabla^2\mathcal{L}(\kappa(s))\right\|ds.
\end{aligned}$$

Invoking $\|\boldsymbol{w}_{t+1} - \boldsymbol{w}_t\| = h(\boldsymbol{g}_t)\|\boldsymbol{g}_t\|$ and $\|\nabla^2 \mathcal{L}(\kappa(s))\| \leq \beta$ we have

$$\mathbb{E}[\mathcal{L}(\boldsymbol{w}_{t+1}) \mid \boldsymbol{w}_t] \leq \mathcal{L}(\boldsymbol{w}_t) - \mathbb{E}[h(\boldsymbol{g}_t)\langle \boldsymbol{g}_t, \nabla \mathcal{L}(\boldsymbol{w}_t)\rangle \mid \boldsymbol{w}_t] + \frac{\beta}{2}\mathbb{E}\left[h(\boldsymbol{g}_t)^2\|\boldsymbol{g}_t\|^2 \mid \boldsymbol{w}_t\right].$$

Substituting $\nabla \mathcal{L}(\boldsymbol{w}_t) + \boldsymbol{g}_t - \nabla \mathcal{L}(\boldsymbol{w}_t)$ for $\boldsymbol{g}_t$ in the second and the third term above, we get

$$\mathbb{E}[\mathcal{L}(\boldsymbol{w}_{t+1}) \mid \boldsymbol{w}_t] \leq \mathcal{L}(\boldsymbol{w}_t) - \mathbb{E}[h(\boldsymbol{g}_t)\langle \boldsymbol{g}_t - \nabla \mathcal{L}(\boldsymbol{w}_t), \nabla \mathcal{L}(\boldsymbol{w}_t)\rangle \mid \boldsymbol{w}_t] - \mathbb{E}[h(\boldsymbol{g}_t) \mid \boldsymbol{w}_t]\|\nabla \mathcal{L}(\boldsymbol{w}_t)\|^2$$
$$+ \frac{\beta}{2}\mathbb{E}\left[h(\boldsymbol{g}_t)^2\left(\|\nabla \mathcal{L}(\boldsymbol{w}_t)\|^2 + \|\boldsymbol{g}_t - \nabla \mathcal{L}(\boldsymbol{w}_t)\|^2 + 2\langle \nabla \mathcal{L}(\boldsymbol{w}_t), \boldsymbol{g}_t - \nabla \mathcal{L}(\boldsymbol{w}_t)\rangle\right) \mid \boldsymbol{w}_t\right].$$

Recalling that $\eta\delta \leq h(\boldsymbol{g}_t) \leq \eta$ and given that $\delta \in (0,1)$,

$$\mathbb{E}[\mathcal{L}(\boldsymbol{w}_{t+1}) \mid \boldsymbol{w}_t] \leq \mathcal{L}(\boldsymbol{w}_t) - \mathbb{E}[h(\boldsymbol{g}_t)\langle \boldsymbol{g}_t - \nabla \mathcal{L}(\boldsymbol{w}_t), \nabla \mathcal{L}(\boldsymbol{w}_t)\rangle \mid \boldsymbol{w}_t]$$
$$- \eta\delta\|\nabla \mathcal{L}(\boldsymbol{w}_t)\|^2 + \frac{\beta\eta^2}{2}\|\nabla \mathcal{L}(\boldsymbol{w}_t)\|^2 + \frac{\beta\eta^2\theta^2}{2} + \beta\mathbb{E}\left[h(\boldsymbol{g}_t)^2\langle \boldsymbol{g}_t - \nabla \mathcal{L}(\boldsymbol{w}_t), \nabla \mathcal{L}(\boldsymbol{w}_t)\rangle \mid \boldsymbol{w}_t\right].$$

Now we invoke Lemma A.3 on the second term above and Lemma A.2 on the last term of the RHS above and take total expectations to get,

$$\mathbb{E}[\mathcal{L}(\boldsymbol{w}_{t+1})] \leq \mathbb{E}[\mathcal{L}(\boldsymbol{w}_t)] + \{\eta(1-\delta)\theta + \beta\eta^2\theta\}\mathbb{E}[\|\nabla \mathcal{L}(\boldsymbol{w}_t)\|] - \left(\eta\delta - \frac{\beta\eta^2}{2}\right)\mathbb{E}\left[\|\nabla \mathcal{L}(\boldsymbol{w}_t)\|^2\right] + \frac{\beta\eta^2\theta^2}{2}.$$

Given a $T \in \mathbb{Z}^+$, summing the above over all $t = 1, \ldots, T$ and recalling that $\boldsymbol{w}_1$ is an arbitrary non-random initialization, we have

$$\left(\eta\delta - \frac{\beta\eta^2}{2}\right)\sum_{t=1}^{T}\mathbb{E}\left[\|\nabla \mathcal{L}(\boldsymbol{w}_t)\|^2\right] \leq \mathcal{L}(\boldsymbol{w}_1) - \mathbb{E}[\mathcal{L}(\boldsymbol{w}_{T+1})] + \{\eta(1-\delta)\theta + \beta\eta^2\theta\}\sum_{t=1}^{T}\mathbb{E}[\|\nabla \mathcal{L}(\boldsymbol{w}_t)\|] + \frac{\beta\eta^2\theta^2}{2}T.$$

Invoking the inequality, $\theta \cdot \|\nabla \mathcal{L}(\boldsymbol{w}_t)\| \leq \frac{1}{2}(\theta^2 + \|\nabla \mathcal{L}(\boldsymbol{w}_t)\|^2)$, and that $\mathcal{L} \geq 0$ we get

$$\left(\eta\delta - \frac{\beta\eta^2}{2}\right)\sum_{t=1}^{T}\mathbb{E}\left[\|\nabla \mathcal{L}(\boldsymbol{w}_t)\|^2\right] \leq \mathcal{L}(\boldsymbol{w}_1) + \{\eta(1-\delta) + \beta\eta^2\}\sum_{t=1}^{T}\mathbb{E}[\frac{1}{2} \cdot \|\nabla \mathcal{L}(\boldsymbol{w}_t)\|^2]$$
$$+ \left(\frac{\beta\eta^2 + \eta(1-\delta) + \beta\eta^2}{2}\right)\theta^2 T.$$

The above implies

$$\left(\eta\delta - \frac{\beta\eta^2}{2} - \frac{\eta(1-\delta) + \beta\eta^2}{2}\right)\sum_{t=1}^{T}\mathbb{E}\left[\|\nabla \mathcal{L}(\boldsymbol{w}_t)\|^2\right] \leq \mathcal{L}(\boldsymbol{w}_1) + \left(\frac{2\beta\eta^2 + \eta(1-\delta)}{2}\right)\theta^2 T.$$

Invoking the assumption that $\delta > \frac{\left(1+\left(\frac{\epsilon}{\theta}\right)^2\right)}{\left(1+3\left(\frac{\epsilon}{\theta}\right)^2\right)} > \frac{1}{3}$ and $\eta < \frac{\delta\left(1+3\left(\frac{\epsilon}{\theta}\right)^2\right) - \left(1+\left(\frac{\epsilon}{\theta}\right)^2\right)}{2\beta\left(1+\left(\frac{\epsilon}{\theta}\right)^2\right)} < \frac{3\delta-1}{2\beta}$ we get

$$\min_{t=1,\ldots,T}\mathbb{E}\left[\|\nabla \mathcal{L}(\boldsymbol{w}_t)\|^2\right] \leq \frac{1}{T}\sum_{t=1}^{T}\mathbb{E}\left[\|\nabla \mathcal{L}(\boldsymbol{w}_t)\|^2\right] \leq \frac{\mathcal{L}(\boldsymbol{w}_1)}{T \cdot \left(\frac{\eta}{2}(3\delta-1) - \beta\eta^2\right)} + \left(\frac{2\beta\eta^2 + \eta(1-\delta)}{2 \cdot \left(\frac{\eta}{2}(3\delta-1) - \beta\eta^2\right)}\right)\theta^2.$$

Now for an arbitrary $\epsilon > 0$, we can solve the inequality

$$\frac{\eta(1-\delta) + 2\beta\eta^2}{\eta(3\delta-1) - 2\beta\eta^2} < \left(\frac{\epsilon}{\theta}\right)^2$$

and thus we get,

$$\eta \in \left(0, \frac{\delta\left(1 + 3\left(\frac{\epsilon}{\theta}\right)^2\right) - \left(1 + \left(\frac{\epsilon}{\theta}\right)^2\right)}{2\beta(1 + \left(\frac{\epsilon}{\theta}\right)^2)}\right).$$

Note that the above upper bound on $\eta$ is the range of $\eta$ chosen in the statement. Thus we have established the claimed theorem statement. $\square$

### A.2  Proof of Theorem 2.2

*Proof.* With the given choices of $\eta, \delta$ and $\beta$ as in the theorem statement, we have

$$\frac{1}{\eta \cdot \left(\frac{3\delta-1}{2} - \beta\eta\right)} = \frac{16(1+\epsilon'^2)^2(1+3\epsilon'^2)}{\epsilon'^2(3\epsilon'^4 + 9\epsilon'^2 + 4)}$$

$$= \frac{4}{\epsilon'^2} + 11 + \frac{\epsilon'^2}{4} + \frac{51\epsilon'^4}{16} + \mathcal{O}(\epsilon'^6).$$

Substituting the above into the guarantee of Theorem A.1, meaning the RHS of (20), along with using $T \geq 1/\epsilon'^4$, we get the result claimed. $\square$

