# OpenReview forum: "Regularized Gradient Clipping Provably Trains Wide and Deep Neural Networks"
_TMLR — Accepted by TMLR_

### Review · Reviewer_xetK · 2025-04-11

**Summary Of Contributions:**

This paper introduces $\delta$-GClip, an algorithm for stabilizing training by combining gradient clipping with a regularization term. The key idea is to allow gradients to be clipped more smoothly, rather than abruptly, by framing clipping as a regularized optimization problem. The authors present a unified perspective on common clipping methods (like norm-based and adaptive clipping), show theoretical stability properties of $\delta$-GClip, and demonstrate improved performance on several tasks including large-scale vision models and reinforcement learning.

**Audience:**

Yes

**Claims And Evidence:**

Yes

**Requested Changes:**

More systematic analysis or guidance on how to set the regularization parameter across tasks will strengthen the case for the proposed method as a practical, low-effort improvement.

It would also be helpful to add more comparisons to recent clipping or optimization strategies in large-scale training and reinforcement learning to reinforce the empirical claims.

Optional:
- More intuitive explanations or diagrams to clarify the optimization interpretation of $\delta$-GClip, making it more accessible.
- Show whether $\delta$-GClip also affects generalization at late training stages or primarily helps early convergence.
- Include more ablation studies to separate the effects of $\delta$-GClip from other training details (e.g., learning rates or model sizes).

**Strengths And Weaknesses:**

## Strengths
- The paper offers a clean, unified view of gradient clipping approaches through the lens of regularized optimization, which is both theoretically grounded and intuitive.
- The method is simple and incurs little overhead, which makes it likely to be adopted in practice.
- Strong experimental evidence shows that  $\delta$-GClip improves convergence speed and robustness across a range of settings: ViTs, diffusion models, and policy gradients in RL.

## Weaknesses
- The paper could better contextualize its theory in relation to other robust optimization or clipping strategies, including, for example, connections to trust region methods, proximal point methods, or more modern noise-aware optimizers could further clarify where $\delta$-GClip fits in the broader optimization landscape.
- While the method is said to be robust, the paper could more clearly communicate how sensitive $\delta$-GClip is to the choice of the regularization coefficient.
- In the reinforcement learning experiments, the comparison set is relatively narrow. Many RL algorithms already include sophisticated stabilization mechanisms (e.g., the clip function in PPO, trust regions in TRPO, or gradient penalty methods). Without direct comparisons to these or an integration of $\delta$-GClip into such methods, it's unclear how much of the improvement is due to $\delta$-GClip versus general tuning or architectural choices.

---

> ### Author Response · Authors · 2025-05-22
> **Response**
>
> We would like to thank the reviewer for their critical reviewing of our paper.
> In the following we would like to offer responses to the key issues raised.
>
> > More systematic analysis or guidance on how to set the regularization parameter across tasks will strengthen the case for the proposed method as a practical, low-effort improvement.
>
> From all our experiments presented so far, we can see that setting the $\delta$ parameter $\sim 10^{-6}$ gives stable performance in almost all scenarios - particularly for transformers. Further, in cases where a baseline result might be available for performance of standard clipping, the $\eta$ and the $\gamma$ parameter of that can be a good estimate of setting the same for $\delta-$GClip. We posit that a small amount of hyperparameter search around these values should typically lead to good parameter choices for $\delta-$GClip.
>
> > It would also be helpful to add more comparisons ... diffusion models, reinforcement learning to reinforce the empirical claims.
>
> We would like to submit that diffusion models and reinforcement learning are outside the scope of this work --- we have made no claims about our algorithm in those contexts. Nothing in our paper corresponds to diffusion models or RL.
>
>
> > More intuitive explanations or diagrams to clarify the optimization interpretation of $\delta-$GClip, making it more accessible.
>
> Kindly note the new equation 1 that we have now written into the file --- which makes the algorithm more intuitive. In here we have presented our idea as a particular gradient-norm dependent choice of step-lengths for GD. Equation 1 makes it explicit that our $\delta-$GClip chooses $3$ kinds of step-lengths for GD, $\eta$ when gradient norms are less than $\gamma$, $\eta \delta$ when gradient norms are larger than $\frac{\gamma}{\delta}$ and inversely proportional to the gradient norm for the intermediate range.
>
>
> > show whether $\delta$-GClip also affects generalization at late training stages or primarily helps early convergence.
>
> As can be seen from all the figures given, we have no reason to believe that $\delta-$GClip has any significant change in influence between early and late times of training. Ofcourse, we do see that in the sporadic occasions when the ``max'' argument in our step-size returns $\delta$, its during the early stages training when the gradient norms are typically high for standard nets.
>
> > Include more ablation studies to separate the effects of $\delta-$GClip from other training details (e.g., learning rates or model sizes)
>
> To address this query we focus on our studies on the transformers, which are our most difficult demonstration cases. Kindly note the new tabulated results for transformer experiments -- Table 1 for studies on Vision Transformers and Table 2 for studies on BERT variants. These show the attained training loss and test accuracy for all such experiments with means and standard deviations over repeats of the experiment. We have now reported these training statistics over multiple transformers and data and for multiple optimizer choices for each. By reporting alongside the data for Adam on the same setups, we have given a robust demonstration of the performance of our optimizer.

---

### Review · Reviewer_rGm3 · 2025-04-30

**Summary Of Contributions:**

The authors strive to introduce delta-gradient clipping, a regularized gradient clipping method that guarantees convergence to global minima in deep neural networks with squared loss, given sufficient layer width. Compared to conventional approaches, the core of the proposed method is to add a lower bound during the gradient clipping process. Some empirical results show delta-gradient clipping competes with current methods, including transformer architectures.

**Audience:**

Yes

**Broader Impact Concerns:**

I have not found any discussions about the limitations and potential negative societal impact. But in my opinion, this may not be a problem, since the work only focuses on the optimization in deep learning. Still, it is highly encouraged to add corresponding discussions.

**Claims And Evidence:**

No

**Requested Changes:**

See Weakness

**Strengths And Weaknesses:**

**Strengths**

1. The paper is clearly written and easy to follow.
2. Analyzing the convergence behavior is a significant area of interest in optimization, particularly in the context of scalable deep neural networks.
3. The authors offer a rigorous theoretical analysis of the convergence properties of the proposed delta-gradient clipping algorithm, providing a solid foundation for understanding its behavior.

**Weakness**

1. In my view, the introduction does not well articulate the motivation and significance of this work. For example, it remains unclear why the authors emphasize the convergence analysis of Adam optimization, given that the paper does not provide further theoretical contributions on this topic. Additionally, the problem that the proposed delta-gradient clipping method aims to address should be explicitly stated. In other words, what motivates the authors to propose this method? I could understand what the authors would like to contribute, but the writing here is not quite clear.

    And by the way, several papers have focused on analyzing the convergence of Adam under certain assumptions. The authors should properly acknowledge and discuss these existing results rather than claiming, "However, to the best of our knowledge, there has not been so far a theoretical guarantee for any adaptive gradient algorithm to converge to the global minima of deep neural network loss surfaces." This statement requires revision to accurately reflect the literature.

    I recommend that the authors revise the introduction to more clearly articulate the motivation and objectives of the paper, as well as the significance of the proposed approach. At present, it lacks clarity regarding the core contributions and the specific problems the authors aim to solve.

2. The core of the proposed method, i.e. delta-gradient clipping, is to add a lower bound during the gradient clipping process, as compared to conventional approaches. In my view, incorporating such a lower bound does not appear to offer substantial practical benefits for training. Furthermore, the convergence analysis largely follows the framework established in the paper "Loss landscapes and optimization in over-parameterized non-linear systems and neural networks", and thus the results presented are largely anticipated.

3. The presentation of empirical results should be especially improved. At present, the figures are difficult to interpret, and the visual clarity is insufficient. At a minimum, a table regarding the comparisons between the proposed method and the baselines should be provided. Also, the experiments should involve several different architectures on Cifar-{10, 100}, and ImageNet.

    Moreover, an ablation study w.r.t the extra hyper-parameters delta should be provided. Currently, delta is set to a very small value, and it remains unclear whether the gradient norm actually reaches this lower bound during training. The authors should provide evidence or analysis to clarify whether the proposed lower bound is actively engaged and contributes to training dynamics.

---

> ### Author Response · Authors · 2025-05-01
> **A quick clarification request**
>
> Dear Reviewer,
>
> Many thanks for reviewing our work and for sharing thoughtful feedback.
>
> While we prepare a response to your review, we wanted to ask clarification regarding this:
>
> > And by the way, several papers have focused on analyzing the convergence of Adam under certain assumptions. The authors should properly acknowledge and discuss these existing results rather than claiming, "However, to the best of our knowledge, there has not been so far a theoretical guarantee for any adaptive gradient algorithm to converge to the global minima of deep neural network loss surfaces." This statement requires revision to accurately reflect the literature.
>
> Of course we agree on the importance of citing the related literature. However, to avoid a "shoot and miss" situation, so to speak, we'd appreciate if you could clarify what specific literature items you'd like us to include in the revised version. Anticipated thanks!
>
> All the best,
>
> The Authors

---

> ### Author Response · Authors · 2025-05-22
> **Response**
>
> We would like to thank the reviewer for their critical reviewing of our paper.
> In the following we would like to offer responses to the key issues raised.
>
> We begin the paper discussing Adam because it and its variants are the most popular kinds of successful deep-learning heuristics. The third paragraph of the introduction emphasizes that there are no known convergence guarantees on deep-neural nets for any kind of adaptive gradient algorithm. And as the last/italicized line just before the summary section emphasizes, that *we close this critical gap by giving a first-of-its-kind deep learning algorithm that is of practical benefit while being rigorously provable to be minimizing loss functions on deep neural nets.* And our method is also adaptive.
>
> We would like to point out that our Section 4, ''Related Works'' includes  dedicated sections ''Literature Review of Theory for Adam''  and ``Review of Theory for Adaptive Gradient Methods Training Neural Nets'' where we have surveyed the best results we are aware of to bolster the statement in the introduction that such algorithms have no known deep-learning guarantees.
>
> Kindly let us know if we have missed any critical reference and we could then include a discussion for that too.
>
> > incorporating such a lower bound does not appear to offer substantial practical benefits for training
>
> The introduction of the $\delta > 0$ parameter is critical to how the proof works and to the best of our knowledge there is no known way to prove convergence on neural losses of the famously successful standard gradient clipping ($\delta = 0$ for our $\delta-$GClip). It is anticipated (and demonstrated in our work) that the introduction of a small $\delta$ parameter would only have marginal influence on performance metrics as compared to standard GClip.
>
> But the innovation of introducing this infinitesimal parameter $\delta$ has theoretical significance as it opens up a path towards theoretical foundations for gradient clipping. Given that the proof of convergence presented here works at an arbitrarily small $\delta >0$, our formalism can be seen as a novel contribution towards explaining the efficacy of gradient clipping.
>
> The experiments show the performance of $\delta-$GClip to be at par with SOTA heuristics of deep-learning, even for transformers -- Table 1 and 2.
>
> Thus we have presented a first-of-its-kind instance of a deep-learning algorithm that provably trains nets while being practically competitive.
>
> > the figures are difficult to interpret, and the visual clarity is insufficient. At a minimum, a table regarding the comparisons between the proposed method and the baselines should be provided.
>
> We updated the figures of the transformer experiments to improve their visual clarity by increasing the color contrast. We have also added table 1 and 2 to summarize the comparison results between Adam and our method across different transformer architectures.

---

### Review · Reviewer_ftSa · 2025-05-08

**Summary Of Contributions:**

The paper studies convergence of a gradient-clipped version of SGD in wide neural networks for the squared loss. The proof of the main result is based on the PL* condition, which has been proven itself by Liu et al recently, utilising the NTK of the model. Furthermore, experiments are presented that complement the theoretical analysis and demonstrate the applicability of the algorithm for training neural networks.

**Audience:**

Yes

**Broader Impact Concerns:**

None.

**Claims And Evidence:**

Yes

**Requested Changes:**

I believe the paper is interesting and ready for publication at its current form. However, the authors could consider improving the papers along the following axes:

- I believe the statement of Theorem 2.1 and the subsequent discussion could be improved. In particular, the last two sentences of the Theorem's statement appear to be misphrased. Also, the authors could perhaps discuss the smoothness assumptions for F(w).
- Similarly, the smoothness of L is assumed for Theorem 2.2, yet the authors do not explicitly comment on it.
- On the bottom of pg. 4, there is a typo: "deep-learnning"
- In the experiments section: In Figure 2, the clipped algorithms display a non-monotonic behaviour in the loss, before epoch 150 when an additional learning rate reduction is applied. Can you comment on it? Do you have any intuition on why it happens?
- More random seeds for the experimental study.

**Strengths And Weaknesses:**

The paper has several strengths:

- It is well written and it presents clearly its contributions.
- It provides an optimization result in the case of a non-trivial adaptive algorithm. The authors also consider the stochastic case. The proofs appear to be correct.
- It contains an extensive experimental study with neural networks applied in several applications.

However, I identified two weakness:

- Admittedly, the technical contribution of the paper is limited; the main tool of the paper comes from the PL* condition for wide networks.
- The paper claims competitiveness of the proposed algorithm with SGD and Adam, yet all the results are with respect to one random seed. I would recommend being more cautious with some claims in the paper regarding the experiments, as limited conclusions can be drawn from single-seed experiments.

---

> ### Author Response · Authors · 2025-05-22
> **Response**
>
> We would like to thank the reviewer for their critical reviewing of our paper.
> In the following we would like to offer responses to the key issues raised.
>
> Firstly, we note that the mentioned typos have been fixed.
>
>
> >..the statement of Theorem 2.1 and the subsequent discussion could be improved.
>
> In the revised draft we have now improved the language of the statement of Theorem 2.1.
>
> > ...discuss the smoothness assumptions for F(w). Similarly, the smoothness of L is assumed for Theorem 2.2
>
> We note that such a Lipschitz smoothness assumption for the loss function is standard for such convergence proofs. In this particular case this smoothness of the training loss is inherited from the activations of the net having been assumed to be $\beta_\sigma$ smooth, as stated in Definition 4. This is immediately seen to be true for $\tanh$ and sigmoid nets.
>
> > In Figure 2, the clipped algorithms display a non-monotonic behaviour in the loss, before epoch 150 when an additional learning rate reduction is applied.
>
> Firstly, we note that this phenomenon you pointed out is for the GClip algorithm which corresponds to setting $\delta =0$ for our algorithm. We recall that there is no known neural net for which the standard GClip algorithm provably converges and thats what motivates our proposed innovation. Hence, we posit that oddities in the behaviour of GClip are possibly reflecting why its proof -- if it exists --- cannot go via the proof techniques used in this work.
>
> > More random seeds for the experimental study.
>
> To address this query we focus on our studies on the transformers, which are our most difficult demonstration cases. Kindly note the new tabulated results for transformer experiments -- Table 1 for studies on Vision Transformers and Table 2 for studies on BERT variants. These show the attained training loss and test accuracy for all such experiments with means and standard deviations over repeats of the experiment. We have now reported these training statistics over multiple transformers and data and for multiple optimizer choices for each. By reporting alongside the data for Adam on the same setups, we have given a robust demonstration of the performance of our optimizer.

---

> > ### Comment · Reviewer_ftSa · 2025-05-27
> > **Response**
> >
> > Thank you for your reply. My concerns have been addressed.

---

### Comment · Editors_In_Chief · 2025-07-04
**Post-Publication Edit**

On July 3, 2025, at the request of the authors, the Editors-in-Chief uploaded a new camera ready version. This includes a number of minor wording changes to improve clarity. Additionally, the ordering of the third and fourth authors was swapped (which all authors individually consented to via email).

---

### Decision · Action_Editor_7pMb · 2025-06-12

**Recommendation:** Accept as is

**Additional Comments:**

The paper proposes an algorithm based on gradient clipping and a particular step-size scheduling. By leveraging the PL* condition, the authors provide a proof of convergence; the competitiveness of the method w.r.t. SGD and Adam is also argued in numerical experiments.

The three reviewers have praised the clarity of the exposition and have shown appreciation for the novel methodology. Some issues were raised concerning the experimental study (single-seed experiments) and, more generally, about the novelty of the technical contribution (linked to the PL* condition introduced in a different work) and about the impact of the proposed method as compared to conventional approaches. The rebuttal of the authors has resolved the more specific issues by adding more experiments, thus improving the quality of the manuscript.

My opinion is that the strengths of the paper clearly outweigh the more general weaknesses mentioned above. The method is novel and the theoretical analysis solid. Thus, I think this will be a nice addition to TMLR and recommend acceptance.

**Audience:**

Yes

**Audience Explanation:**

The general topic (gradient-based optimization) is certainly of broad interest to TMLR's audience and the specific results, while not entirely surprising at the technical level, will be of interest to a more specialized audience.

**Claims And Evidence:**

Yes

**Claims Explanation:**

The reviewers have praised the solid theoretical insights and the rigorous convergence analysis. The claims are also supported by a number of numerical experiments, some of which were added during the reviewing process. As such, overall the paper supports its claims in a convincing way.

---

> ### Author Response · Authors · 2025-06-18
> **Thanks!**
>
> Thanks a lot for your kind comments and recommendation for accept!
>
> Should we go ahead and submit a camera-ready draft?

---

> > ### Comment · Action_Editor_7pMb · 2025-06-18
> >
> > Dear authors,
> >
> > yes, you should submit your camera ready.
> >
> > Thank you for your nice contribution to TMLR,
> >
> > Action Editor

---

> > > ### Author Response · Authors · 2025-06-19
> > > **Final Version**
> > >
> > > Thanks for the comments.
> > >
> > > Kindly note that we have now uploaded the final version.